# Ecological Adaptation of Two Dominant Conifer Species to Extreme Climate in the Tianshan Mountains

Xuan Wu [1,2], Liang Jiao [1,2,*] , Xiaoping Liu [1,2], Ruhong Xue [1,2], Changliang Qi [1,2] and Dashi Du [1,2]

1 College of Geography and Environment Science, Northwest Normal University, No.967, Anning East Road, Lanzhou 730070, China; 2020212684@nwnu.edu.cn (X.W.); 2018212281@nwnu.edu.cn (X.L.); 2022120260@nwnu.edu.cn (R.X.); 2019212378@nwnu.edu.cn (C.Q.); 2020212664@nwnu.edu.cn (D.D.)
2 Key Laboratory of Resource Environment and Sustainable Development, Oasis Northwest Normal University, Lanzhou 730070, China
* Correspondence: jiaoliang@nwnu.edu.cn

**Abstract:** With global warming, the frequency, intensity, and period of extreme climates in more areas will probably increase in the twenty first century. However, the impact of climate extremes on forest vulnerability and the mechanisms by which forests adapt to climate extremes are not clear. The eastern Tianshan Mountains, set within the arid and dry region of Central Asia, is very sensitive to climate change. In this paper, the response of *Picea schrenkiana* and *Larix sibirica* to climate fluctuations and their stability were analyzed by Pearson's correlation based on the observation of interannual change rates of climate indexes in different periods. Additionally, their ecological adaptability to future climate change was explored by regression analysis of climate factors and a selection of master control factors using the Lasso model. We found that the climate has undergone significant changes, especially the temperature, from 1958 to 2012. Around 1985, various extreme climate indexes had obvious abrupt changes. The research results suggested that: (1) the responses of the two tree species to extreme climate changed significantly after the change in temperature; (2) Schrenk spruce was more sensitive than Siberian larch to extreme climate change; and (3) the resistance of Siberian larch was higher than that of Schrenk spruce when faced with climate disturbance events. These results indicate that extreme climate changes will significantly interfere with the trees radial growth. At the same time, scientific management and maintenance measures are taken for different extreme weather events and different tree species.

**Keywords:** ecological adaptability; resistance indexes; lasso model; future prediction; tree-ring; climate change

## 1. Introduction

Forest ecosystems are an important part of terrestrial ecosystems. Plants form a long-term, relatively stable and dynamic balance with the regional climate during their growth [1]. The occurrence of extreme climatic events has strongly disturbed the inherent balance between trees and environmental factors, causing significant changes in the phenological activities, productivity levels, and community succession statuses of forest ecosystems and thereby increasing community survival pressure and the potential risk of local extinction [2–4].

In the context of worldwide warming, the frequency, intensity, and period of maximum weather events still increase, and the impacts of climate on tree growth are growing stronger [5,6]. Therefore, it is particularly important to study the response relationship between extreme climate indicators and tree growth in order to explore tree growth strategies and formulate management and protection measures under future climate change. At present, in order to further strengthen forest management and protection and scientific planning, climate change modeling tools such as the Forest Landscape Model (RCMs), which includes the effects of climate and management on forest dynamics, have been

used to predict the layout of forest growth under future climate impacts, especially in forest areas with complex terrain [7]. At low latitudes in Bangladesh, the radial growth of *Chukrasia tabularis* was reduced by 54% and 48.7% by the two extreme drought events in 1999 and 2006, respectively [8]. At an identical time, extreme precipitation events are more likely to flood plants in tropical regions due to abundant water and heat resources, which will lead to hypoxia of plant roots, reduced respiration, and eventually plant death [9]. At middle and high latitudes in the virgin forests of southern Europe, the sensitivity of *Picea abies* and *Abies alba* to climatic and non-climatic disturbances increased due to atmospheric warming and increased drought [10]. Compared to those in low-latitude regions, trees in mid- and high-latitude regions were much more at risk of extreme drought and freezing damage throughout the growing season [11]. Although the impact of extreme precipitation on tree growth is not as direct and significant as the impacts of extreme high-temperature and drought events, studies have shown that extreme precipitation events affect forest carbon sinks and ecohydrological processes, thereby affecting the function and structure of entire forest ecosystems [12–14].

Tree ring width is an important indicator for recording climate change, and its change can truly and objectively reflect the impact of climate change on trees and the ecological response of trees to climate change due to its high resolution, easy preservation, and long time series [15,16].

Tree rings will appear with narrower, false, and missing rings when there are sudden and dramatic variations in climate, such as extreme high temperature, high-intensity precipitation, and frost [17,18]. Specifically, these growth–climate relationships will significantly change with a continuous decrease in the number of cold nights and continuous increases in the frequency and intensity of drought [19,20]. For example, the development of unstable responses of tree growth to temperature change has been documented worldwide [21–24], and therefore, the responses of various tree species to temperature change are clearly varied thanks to variations in physiological and ecological thresholds [9,25,26].

Ecological adaptability refers to the adjustment and recovery abilities of forest ecosystems in the face of extreme climatic conditions. Resistance indexes have been widely used to assess the dynamic changes of forests in response to extreme environmental disturbance events in recent years based on tree rings [27,28]. In addition, a great number of studies have found that differences in the resistance of trees to extreme weather events are affected by many factors, such as age, tree species, and forest stand [23,29–31]. *Fagus sylvatica* in southern Germany exhibited low resistance but high resilience to extreme frost events [32]. In addition, the resistance and resilience of trees are not only related to tree species but are also related to the types of extreme weather events, according to an analysis of the resistance differences of the five main European tree species to spring frost and summer drought [33].

The increase in temperature at the mid-high latitudes of the Northern Hemisphere is considerably beyond the worldwide average attributable to the magnification of the Arctic [34,35]. The forest ecosystems within the arid and dry regions of Central Asia are particularly sensitive and at risk of global climate change thanks to the tough environmental conditions in these regions [3].

As the largest mountain range in Central Asia, the Tianshan Mountains are the origin of most inland rivers and support oasis development in Central Asia. The forest ecosystems in the Tianshan Mountains, with water conservation functions, are of great significance to the natural ecological environment and socioeconomic development in Central Asia [36]. To date, research on tree growth and climate relationships in the Tianshan Mountains has mainly focused on climate reconstruction and conventional climate responses [37,38]. In these studies, the reconstruction of climate indicators using accurate long-sequence tree-ring data is conducive to revealing long-term climate dynamics in different regions of the world, and exploring the response relationship between tree growth and climate is conducive to predicting forest growth patterns under future climate change. However, the effect of extreme climate on forest vulnerability and the mechanisms of forest response

and resistance to extreme climate remain unclear. Therefore, we constructed models of the relationship between radial growth of the dominant tree species (Schrenk spruce and Siberian larch) and extreme climate factors in the eastern Tianshan Mountains, an area with more arid conditions. We aimed to compare the responses and resistances of the two tree species to extreme climate.

## 2. Materials and Methods

### 2.1. Site Description

The study area is found on Barkol Mountain in the eastern Tianshan Mountains (Figure 1). The Barkol region incorporates a typical temperate continental climate with an outsized annual temperature variation. The annual mean temperature is 2.02 °C and presents a significant increasing trend of 0.611 °C/10a ($p < 0.001$), and the annual total precipitation is 220.33 mm, showing an increasing trend of 9.6 mm/10a ($p < 0.05$). The increasing trend of temperature is more obvious than that of total precipitation (Figure 2). The results of the Pettitt test show that the annual mean temperature in the Barkol area underwent a significant abrupt change in 1985 ($p < 0.001$) (Figure 2).

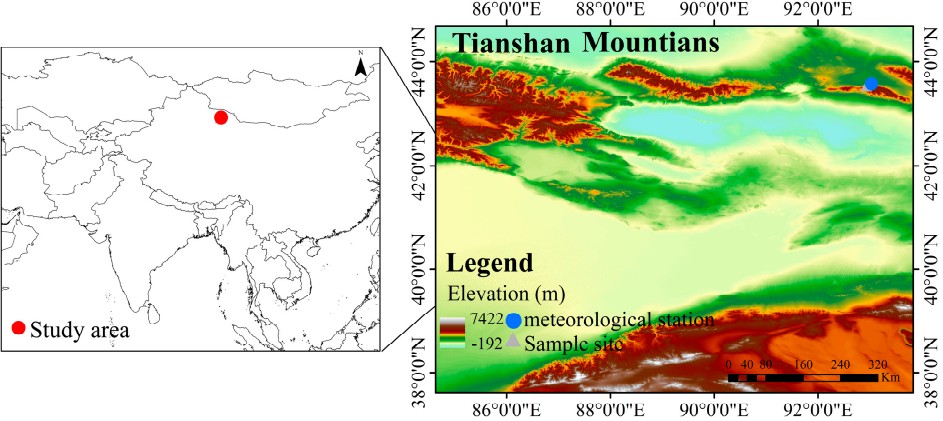

**Figure 1.** Locations of the sampling sites and the nearest meteorological station.

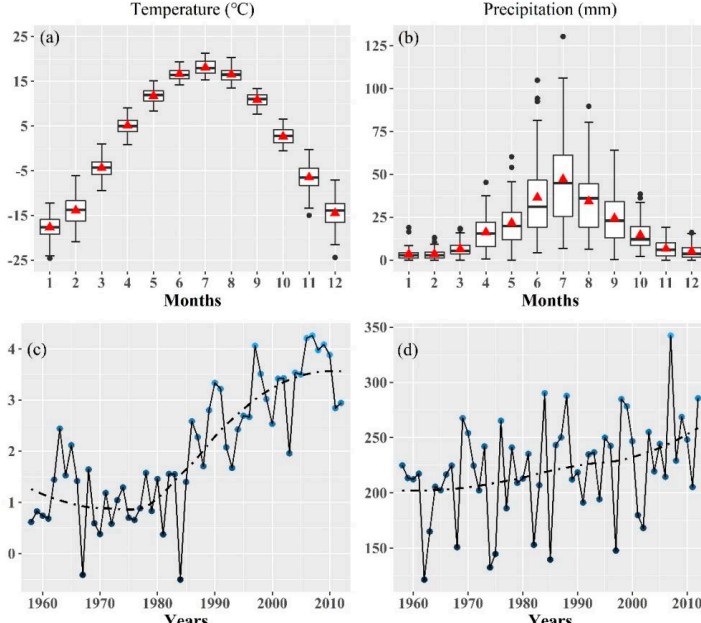

**Figure 2.** Monthly (**a**,**b**) and interannual (**c**,**d**) variation trends of mean temperature and annual total precipitation at the Barkol meteorological station in the eastern Tianshan Mountains from 1958 to 2012.

In panels a and b, the black dots are outliers; red triangles are average values; lower dash is the minimum (excluding outliers); the bottom of the box demarcates 25% of the data; the middle line of the box demarcates 50% of the data (median); the top of the box demarcates 75% of the data; and the upper dash is the maximum (excluding outliers). The dotted line in panels (c) and (d) indicates the interannual change trend of climate elements according to local weighted regression.

## 2.2. Tree Ring Sample Collection and Processing

The sampling sites are settled within the original coniferous mixed forest close to the upper forest line on the northeastern slope of Barkol Mountain with an altitude of roughly 2550–2600 m (Table 1). The sampling sites were less affected by human activity than surrounding areas, with relatively low forest canopy closure, large spacing between trees, and less influence of density. In August 2013, a total of 120 core samples were collected. After damaged and unidentifiable samples were excluded, there remained 60 and 58 sample cores of two species of trees.

**Table 1.** Sampling information of two species of trees.

| Species | Schrenk Spruce | Siberian Larch |
|---|---|---|
| Altitude (m) | 2552 | 2590 |
| Latitude | 43°32.100′ N | 43°32.085′ N |
| Longitude | 92°56.329′ E | 92°56.662′ E |
| Slope | 27° | 33° |
| Aspect | Northeast (40° from North) | North |
| Canopy closure | 0.3 | 0.2 |
| Average tree spacing (m) | 3 | 4.5 |
| Average breast diameter (cm) | 32.4 | 33.8 |
| Average Tree height (m) | 12.3 | 9.7 |
| Average Crown width (m) | 2.7 | 3.7 |

In the laboratory, the core samples of the two tree species were air-dried naturally, fixed in a wooden sample tank with latex, and sanded with mesh sandpapers until the tree rings were visible and clear. First, we used the LINTAB measurement system to measure the ring width of each core with a resolution of 0.001 mm, and then we used the COFECHA program to determine the quality of the cross-dating [39]. Finally, the standard chronology of the two species was obtained by ARSTAN procedure [40].

## 2.3. Chronological Statistical Parameter Calculation

The following statistical parameters of the chronologies, which could be used to assess the reliability of the chronologies, were calculated through the ARSTAN program (Table 2). The SD represents the fluctuation of the tree rings, MS reflects the richness of climate information contained in the chronological sequences, AC1 indicates the influence degree of the previous year's climate on tree growth, R represents the similarity of the chronological sequences, PC1 represents the common information in the tree core samples, the SNR can indicate the amount of climate and environmental information shared by the chronologies, and the EPS reveals the representativeness of the subsample for the whole sample.

**Table 2.** Statistical parameters of the STD of Schrenk spruce and Siberian larch (1958–2012).

| Statistical Parameters | Schrenk Spruce | Siberian Larch |
|---|---|---|
| Sample depth (core/tree number) | 60/30 | 58/29 |
| Sequence length | 1735–2012 (288) | 1761–2012 (252) |
| Standard deviation (SD) | 0.313 | 0.312 |
| Mean sensitivity (MS) | 0.196 | 0.214 |
| AC1 | 0.681 | 0.712 |
| Correlation coefficient ® | 0.483 | 0.382 |
| Mean correlation among trees (R1) | 0.771 | 0.723 |
| Mean correlation between trees (R2) | 0.475 | 0.366 |
| First principal comment (PC1) | 0.507 | 0.435 |
| Signal-to-noise ratio (SNR) | 36.366 | 14.213 |
| Expressed population signal (EPS) | 0.973 | 0.934 |

*2.4. Meteorological Data and Preprocessing*

The meteorological data were obtained from the Barkol National Alpine Meteorological Station in Xinjiang (93°05′ E, 43°06′ N, 1677.2 m). The data set of main climatic factors during 1958–2012 can be obtained from the China Meteorological Data Network (http://data.cma.cn/, accessed on 18 June 2020).

Twenty-seven climate indexes were obtained through the RClimDex program (Table 3). These indexes were developed by the Expert Group on Climate Change Detection, Monitoring, and Indexes (ETCCDI) and the World Meteorological Organization (WMO) cooperative proposal and formulation [41,42]. We further divided the extreme temperature indicators into extreme warmth indicators and extreme cold indicators for better observation and analysis [43]. Among the 27 indexes, 12 included both monthly and annual statistics, and 15 included only annual statistics.

**Table 3.** Extreme climate indexes.

| Type | Class | ID | Indicator Name | Definitions | Unit |
|---|---|---|---|---|---|
| Extreme temperature indexes | Extreme temperature warm indexes | TXx | Max Tmax | Monthly maximum value of daily maximum temp | °C |
| | | TXn | Min Tmax | Monthly minimum value of daily maximum temp | °C |
| | | TX90p | Warm days | Percentage of days when TX > 90th percentile | d |
| | | TN90p | Warm nights | Percentage of days when TN > 90th percentile | d |
| | | WSDI | Warm spell duration indicator | Annual count of days with at least 6 consecutive days when TX > 90th percentile | d |
| | | SU25 | Summer days | Annual count when TX (daily maximum) > 25 °C | d |
| | | TR20 | Tropical nights | Annual count when TN (daily minimum) > 20 °C | d |
| | Extreme temperature cold indexes | TNx | Max Tmin | Monthly maximum value of daily minimum temp | °C |
| | | TNn | Min Tmin | Monthly minimum value of daily minimum temp | °C |
| | | TX10p | Cool days | Percentage of days when TX < 10th percentile | d |
| | | TN10p | Cool nights | Percentage of days when TN < 10th percentile | d |
| | | CSDI | Cold spell duration indicator | Annual count of days with at least 6 consecutive days when TN < 10th percentile | d |
| | | FD0 | Frost days | Annual count when TN (daily minimum) < 0 °C | d |
| | | ID0 | Ice days | Annual count when TX (daily maximum) < 0 °C | d |
| | Other temperature indexes | GSL | Growing season length | Annual (1st Jan to 31st Dec in NH, 1st July to 30th June in SH) count between first span of at least 6 days with TG > 5 °C and first span after July 1 (January 1 in SH) of 6 days with TG < 5 °C | d |
| | | DTR | Diurnal temperature range | Monthly mean difference between TX and TN | °C |

**Table 3.** *Cont.*

| Type | Class | ID | Indicator Name | Definitions | Unit |
|------|-------|-----|----------------|-------------|------|
| Extreme precipitation indexes | Precipitation frequency indexes | R10 | Number of heavy precipitation days | Annual count of days when PRCP $\geq$ 10 mm | d |
| | | R20 | Number of very heavy precipitation days | Annual count of days when PRCP $\geq$ 20 mm | d |
| | | Rnn | Number of days above nn mm | Annual count of days when PRCP $\geq$ nn mm, nn is user defined threshold | d |
| | | CDD | Consecutive dry days | Maximum number of consecutive days with RR < 1 mm | d |
| | | CWD | Consecutive wet days | Maximum number of consecutive days with RR $\geq$ 1 mm | d |
| | Precipitation magnitude indexes | RX1day | Max 1-day precipitation amount | Monthly maximum 1-day precipitation | mm |
| | | Rx5day | Max 5-day precipitation amount | Monthly maximum consecutive 5-day precipitation | mm |
| | | R95p | Very wet days | Annual total PRCP when RR > 95th percentile | mm |
| | | R99p | Extremely wet days | Annual total PRCP when RR > 99th percentile | mm |
| | | PRCPtot | Annual total wet-day precipitation | Annual total PRCP in wet days (RR $\geq$ 1 mm) | mm |
| | Precipitation intensity index | SDII | Simple daily intensity index | Annual total precipitation divided by the number of wet days (defined as PRCP $\geq$ 1.0 mm) in the year | mm/day |

*2.5. Data Analysis*

2.5.1. Climate Data Analysis

First, one-variable linear regression was used to observe the interannual change rates of the 28 climate indexes (annual mean temperature and 27 extreme climate indexes), and then local polynomial regression was used to fit the climate indexes to further observe local changes in the data. The Pettitt test was then used to test for changes in each climate index sequence. Among all the climate indexes (18) that passed the significance test, 12 indexes displayed change years in the mid-1980s, and 6 displayed change years in the mid-1990s. Observing these six indicators, we found that although the change times detected by the test were in the mid-1990s, the increases (or decreases) detected by the test began in the mid-1980s (Figure 3). Therefore, we set the year of abrupt change to 1985 in all cases.

2.5.2. Analysis of the Relationships between Tree Rings and Climate Factors

The correlation coefficients between each climate index and the STDs of the two tree species before climate change (1958–1985) and after climate change (1986–2012) were analyzed by Pearson's correlation. The stability and climate responses of the two tree species were observed when the climatic conditions of the study area change significantly. The climate data from September of the previous year to October of the current year were selected because the radial growth of trees was restricted not only by the climatic factors of the current year but also affected by those of the previous year [15].

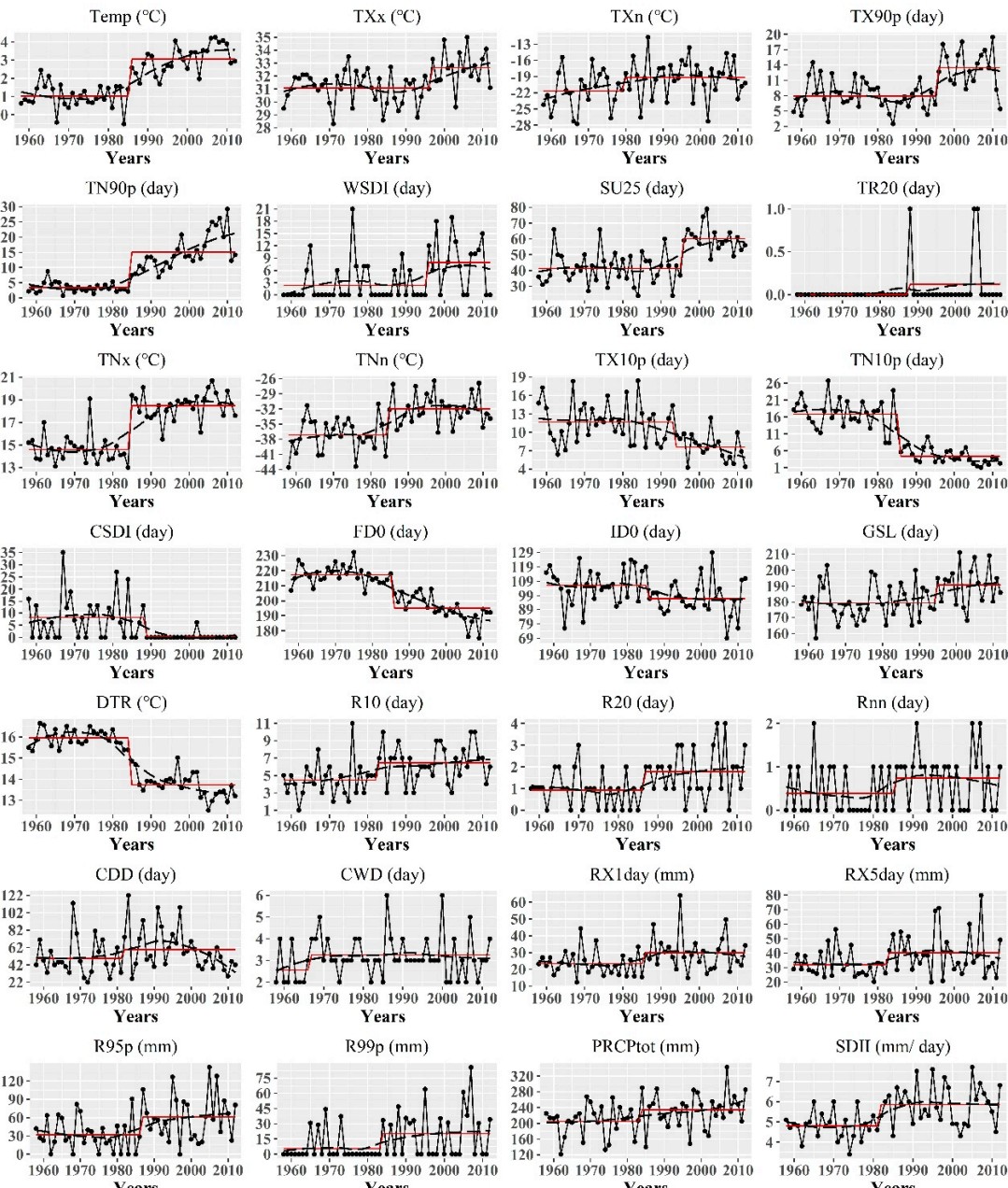

**Figure 3.** Interannual variability and abrupt change of 28 climate indexes in 1958–2012.

### 2.5.3. Pointer Year Selection

We used the STDs to screen the narrow-ring years of the two tree species before and after climate change and used the basal area increment (BAI, cm$^2$/year) for verification. BAI was calculated based on non-standardized raw measurement ring width data as the following equation (Monserud and Sterba, 1996):

$$\text{BAI}_t = \pi\left(r_t^2 - r_{t-1}^2\right)$$

where $r_t$ is a given annual ring corresponding to radial radius at $t$ year and $r_{t-1}$ is a given annual ring corresponding to radial radius at $t-1$ year.

The two tree species generally returned to the growth state observed before the environmental disturbance two years after the appearance of the narrow ring. We chose two years as a window to calculate the 2-year moving average of the chronological sequence to avoid overlap with other disturbance events. In addition, we set two narrow-ring levels

at 80% and 75% of the average of the previous two years and then integrated the screening results of the two chronologies for the tree species. Finally, the years in which both tree species had narrow rings without other narrow rings in the previous two years were selected as pointer years.

2.5.4. Resistance Index Calculation

The resistances of the two tree species to interference events were reflected by their resistance, recovery, and resilience. The calculation formula is as follows (Lloret et al., 2011):

$$RT = TRW_i / TRW_{i-2};$$

$$RC = TRW_{i+2} / TRW_i;$$

$$RS = TRW_{i+2} / TRW_{i-2};$$

where $TRW_i$ represents TRW in the year of environmental stress, $TRW_{i-2}$ represents the average TRW 2 years before the environmental stress, and $TRW_{i+2}$ represents the average TRW 2 years after the environmental stress. RT characterizes the ability of a tree to resist disturbance. RC indicates the degree of recovery of a tree after being disturbed by the environment. RS can measure the growth of a tree after being disturbed and after a period of recovery. Instances of two consecutive drought years were regarded as drought events (for example, 1984 and 1985 in the following text). Afterwards, one-way analysis of variance was used to evaluate the significance of the differences in resistance indexes (RT, RC, and RS) of the two tree species in pointer years before and after the climatic conditions changed.

The above data analyses were all performed in R using the packages "statas", "trend", "car", and "ggplot2".

## 3. Results

### 3.1. Climate Change Characteristics in the Study Area

One-variable linear regression of the 27 climatic factors revealed a very obvious trend of increasing temperature (Figure 3). According to the interannual variations of various extreme climate indicators, the growing season length and extreme temperature warmth index except for tropical night showed a significant increasing trend ($p < 0.05$). The annual diurnal temperature range and the rest of the extreme climate cold indexes showed significant decreasing trends ($p < 0.05$) expect for Max Tmin and Min Tmin. All extreme precipitation indexes showed increasing trends; specifically, the number of heavy precipitation days, number of very heavy precipitation days, very wet days, extremely wet days, and simple daily intensity index showed significant increasing trends ($p < 0.05$).

Pettitt tests of the 27 extreme climate indicators revealed obvious points of abrupt change in the extreme temperature indexes, except for Min Tmax and TR20 ($p < 0.05$) (Table 4). The only extreme precipitation indexes with abrupt change points were R10, R95P, and SDII ($p < 0.05$). The abrupt climate change mainly manifested as a significant increase in temperature, providing evidence that the magnitude and significance of the variation in each precipitation index were smaller than those of the temperature indexes.

**Table 4.** Unary linear regression and Pettit test of 28 climate indexes.

| ID | Unary Linear Regression Equation | R2 | *p* Value of Equation | Mutation Year | *p* Value of Pettitt Test |
|---|---|---|---|---|---|
| Temp | y = 0.3100 + 0.0611x | 0.6200 | $1.3 \times 10^{-12}$ ** | 1985 | $9.3 \times 10^{-9}$ ** |
| TXx | y = 30.700 + 0.0308x | 0.1100 | 0.011 * | 1996 | 0.01202 * |
| TXn | y = −22.100 + 0.0707x | 0.0940 | 0.023 * | 1979 | 0.0843 |
| TX90p | y = 6.5800 + 0.11x | 0.1900 | 0.001 ** | 1995 | 0.0004654 ** |
| TN90p | y = −1.3800 + 0.382x | 0.6900 | $2.9 \times 10^{-15}$ ** | 1984 | $4.415 \times 10^{-9}$ ** |
| WSDI | y = 0.5050 + 0.127x | 0.1200 | 0.0085 ** | 1995 | 0.03846 * |
| SU25 | y = 35.600 + 0.412x | 0.2600 | $6.9 \times 10^{-5}$ ** | 1995 | $4.995 \times 10^{-5}$ ** |
| TR20 | y = −0.0034 + 0.0032x | 0.0490 | 0.1000 | 1987 | 1.501 |
| TNx | y = 13.6000 + 0.1060x | 0.5400 | $1.6 \times 10^{-10}$ ** | 1984 | $3.522 \times 10^{-8}$ ** |
| TNn | y = −38.6000 + 0.1450x | 0.3200 | $5.5 \times 10^{-6}$ ** | 1984 | $1.931 \times 10^{-5}$ ** |
| TX10p | y = 13.50−0 − 0.1170x | 0.2800 | $3.2 \times 10^{-5}$ ** | 1993 | 0.0007737 ** |
| TN10p | y = 21.20−0 − 0.3750x | 0.7000 | $2.4 \times 10^{-15}$ ** | 1985 | $4.002 \times 10^{-9}$ ** |
| CSDI | y = 10.40−0 − 0.2040x | 0.1700 | 0.0016 ** | 1988 | 0.002104 ** |
| FD0 | y = 2−6 − 0.7110x | 0.7200 | $3.4 \times 10^{-16}$ ** | 1985 | $7.16 \times 10^{-9}$ ** |
| ID0 | y = 1−8 − 0.2310x | 0.0770 | 0.0400 * | 1985 | 0.03497 * |
| GSL | y = 175 + 0.2910x | 0.1400 | 0.005 ** | 1994 | 0.02881 * |
| DTR | y = 16.70−0 − 0.0662x | 0.7500 | $1 \times 10^{-17}$ ** | 1984 | $3.232 \times 10^{-9}$ ** |
| R10 | y = 3.8900 + 0.0597x | 0.1700 | 0.0016 ** | 1982 | 0.006723 ** |
| R20 | y = 0.7330 + 0.0212x | 0.1000 | 0.016 * | 1986 | 0.05441 |
| Rnn | y = 0.3660 + 0.0071x | 0.0320 | 0.1900 | 1985 | 0.3658 |
| CDD | y = 53.7000 + 0.0521x | 0.0013 | 0.7900 | 1981 | 0.4919 |
| CWD | y = 3.0100 + 0.00491x | 0.0066 | 0.5600 | 1966 | 0.6511 |
| RX1day | y = 22.5000 + 0.1420x | 0.0560 | 0.0820 | 1985 | 0.06505 |
| RX5day | y = 31.8000 + 0.1710x | 0.0420 | 0.1300 | 1982 | 0.1928 |
| R95p | y = 24 + 0.769x | 0.1200 | 0.0080 ** | 1986 | 0.0333 * |
| R99p | y = 1.8800 + 0.4070x | 0.0960 | 0.0210 * | 1983 | 0.1893 |
| PRCPtot | y = 194 + 0.9600x | 0.1200 | 0.0100 * | 1983 | 0.07576 |
| SDII | y = 4.6100 + 0.0279x | 0.2000 | 0.0006 ** | 1981 | 0.001395 * |

* means significant, ** means extremely significant.

### 3.2. Statistical Parameters of Tree-Ring Width Chronologies for the Two Tree Species

First, the SNR and EPS values of chronologies for Schrenk spruce and Siberian larch were relatively high, indicating that the two chronologies were reliable and suitable for studying the response of tree radial growth to climate change (Table 2). Moreover, the high values of SD, MS, and AC1 also indicated that the chronologies contained much climate information for the two tree species. However, Schrenk spruce exhibited higher consistency between core sequences and was more sensitive to climate change than Siberian larch due to its higher R and PC1 values.

### 3.3. Relationships between Radial Growth and Climatic Factors for the Two Species

The responses of radial growth of the two tree species to climatic factors significantly differed before and after the abrupt temperature change, based on the Pearson's correlation results between the STDs of the two tree species and the annual indexes of 28 climatic factors (Figures 4 and 5). Before the abrupt temperature change, the correlations between Schrenk spruce indexes and the annual climatic factors were poor. However, the STD of Schrenk spruce had a significant negative correlation with summer days (SU25) ($p < 0.05$) and significant positive correlations with cool days (TX10p) and frost days (FD0) ($p < 0.05$) after the abrupt temperature change. The STD of Siberian larch had significant positive correlations with annual mean temperature (Temp), TXn, and TNn ($p < 0.01$) and significant negative correlations with TX10p ($p < 0.01$) before the abrupt temperature change. However, the STD of Siberian larch had a significant positive correlation with FD0 ($p < 0.05$) and a significant negative correlation with R10 ($p < 0.01$) after the abrupt temperature change.

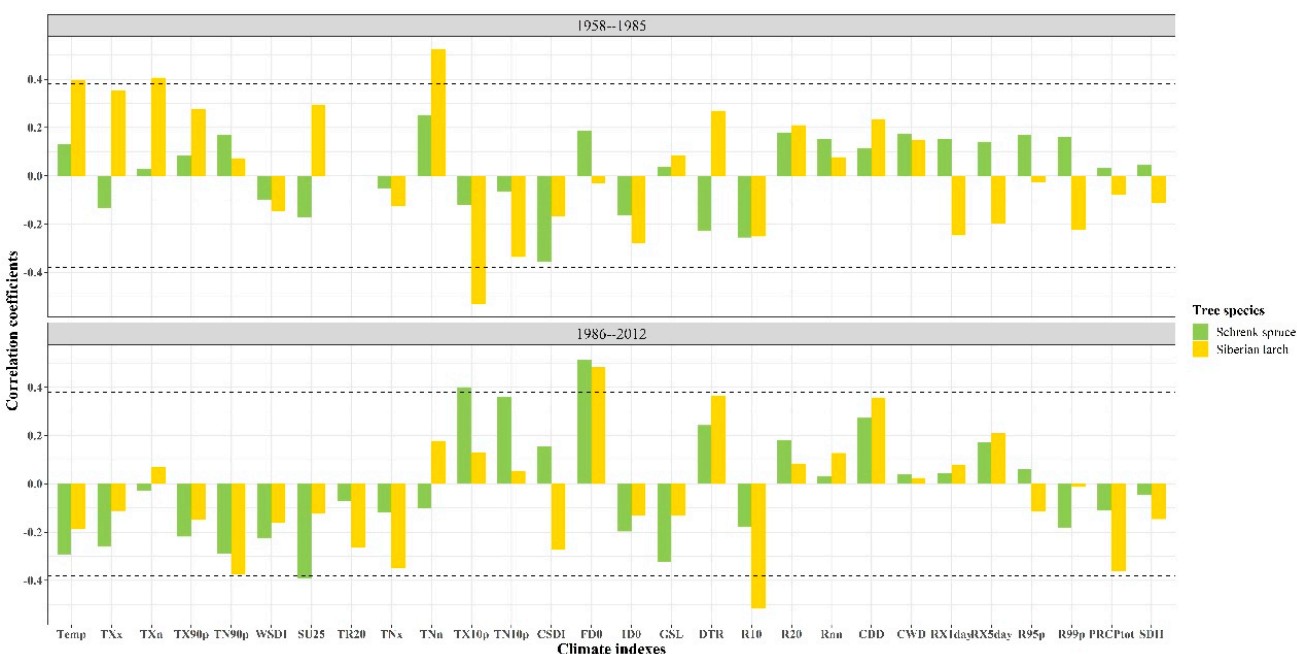

**Figure 4.** Correlation coefficients between the standard chronologies of the two species and annual values of 28 climate indexes during 1958–2012. Dotted line: critical value of significance level *p* < 0.05.

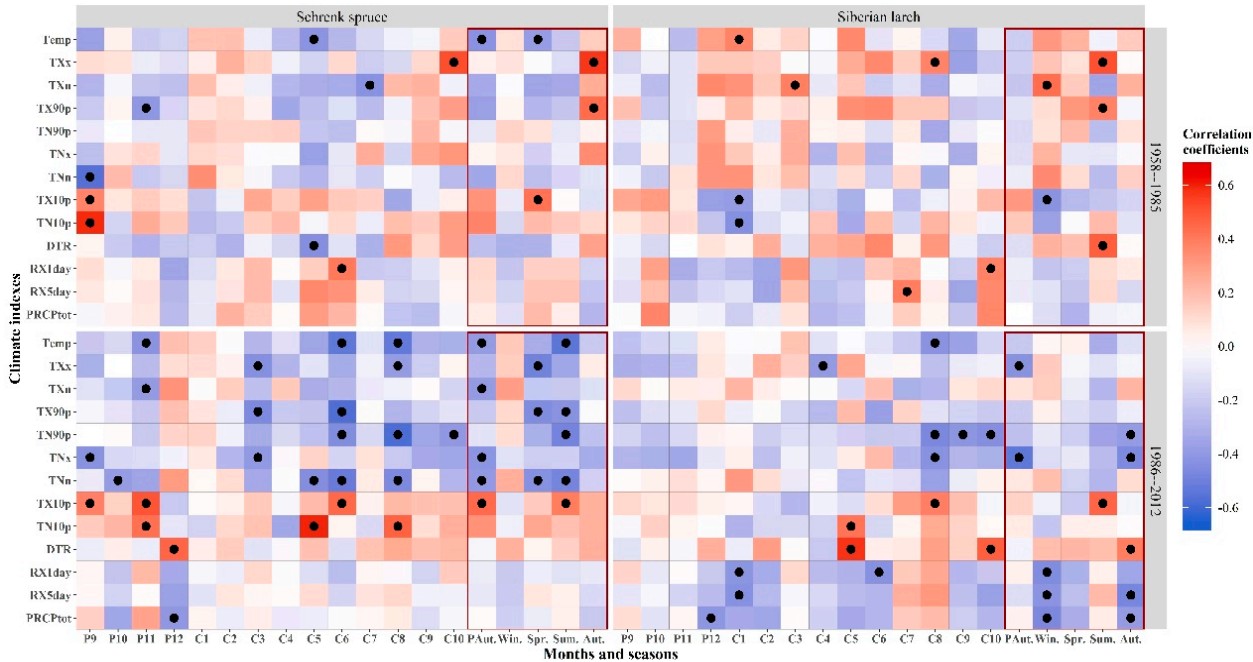

**Figure 5.** Correlation coefficients between the standard chronologies of the two species and monthly values of 13 climate indexes during 1958–2012. Blue: negative correlation, red: positive correlation; black dots: significance level < 0.05, red rectangular box: the correlation coefficients between the standard chronologies of the two species and seasonal climate indicators. P: previous year, C: current year (for example, P9: September of the previous year, C1: January of current year).

The correlation results between the indexes of the two tree species and seasonal factors showed that the STD of Schrenk spruce had a significant correlation with previous autumn temp, spring temp, TX10p, autumn TXx, and TX90p before climate change (1958–1985). That of Schrenk spruce had significant relationships with temp, TXN, TNX, TNN, and TX10P in the autumn of the previous year; TXx, TX90p, and TNn in the spring; and Temp,

TX90p, TN90p, TNn, and TX10p in the summer of the current year after abrupt climate change (1986–2012). The STD of Siberian larch was significantly correlated with only TXn, TX10p in winter, TXx, TX90p, and DTR in summer during the period of 1958–1985. That of Siberian larch had significant correlation with TXX; TNX in the autumn; RX1day, RX5day, and PRCPtot in the winter of the previous year; TX10p in the summer; and TN90p, TNx, DTR, RX5day, and PRCPtot in the autumn of the current year during the period of 1958–1985.

The correlation results between the radial growth of the two tree species and 13 monthly climate indexes from September of the previous year to October of the current year are shown in Figure 5. The STD of Schrenk spruce had significant correlations with P9-TNn (month-climatic factors), P9-TX10p, P9-TN10p, P11-TX90p, C5-Temp, C5-TNx, C5-DTR, C6-RX1day, C7-TXn, and C10-Txx ($p < 0.05$) during the period of 1958–1985. The STD of Schrenk spruce had significant correlations with 25 climatic factors during the period of 1985–2012. The STD of Siberian larch had significant correlations with P10-PRCPtot, P12-TX10p, C1-Temp, C1-TX10p, C1-TN10p, C3-TXn, C7-RX5day, C8-TXx, C10-RX1day, and C10-PRCPtot during the period of 1958–1985. The STD of Siberian larch had significant correlations with 14 climatic factors during the period of 1985–2012. In addition, the negative correlations between the radial growth of Siberian larch and the three precipitation indexes (RX1day, RX5day, and PRCPtot) were significantly enhanced. Schrenk spruce was strongly restricted by the temperatures at the end of the growing season of the previous year and in the early and middle growing seasons of the current year after 1985. However, Siberian larch was mainly affected by the temperatures in the middle and late growing seasons of the current year.

*3.4. Comparison of the Resistance Indexes of the Two Tree Species to Climate Change*

Both Schrenk spruce and Siberian larch were detected in 9 low-value years (Schrenk spruce: 1958, 1974, 1978, 1981, 1995, 1998, 2003, 2008, and 2012; Siberian larch: 1966, 1976, 1981, 1984–1985, 1998–1999, 2003, and 2008) defined as narrow-ring years with a value 20% lower than the average value of the previous two years based on the STDs and BAI, (Figure 6). Schrenk spruce showed only 4 low-value years (STDs: 1974, 2003, 2008, and 2012; BAI: 1998, 2003, 2008, and 2012) and Siberian larch showed 7 low-value years (STDs: 1966, 1976, 1984–1985, 1998–1999, and 2003; BAI: 1966, 1984–1985, 1998–1999, 2003, and 2008) defined as narrow-ring years with a value 25% lower than the average value of the previous two years. The results suggested three years (1981, 1998, and 2003) that were narrow-ring years for both tree species. Since the Siberian larch showed narrow rings in both 1998 and 1999, we chose 1981 and 2003 as pointer years to compare resistance to environmental disturbances between the two tree species.

Figure 7 shows the results of one-way analysis of variance (ANOVA) for the three resistance indicators of the two tree species in 1981 and 2003. In 1981, the resistance indicators of Siberian larch were significantly higher than those of Schrenk spruce (Schrenk spruce: RT = 1.063, RC = 1.06, and RS = 1.128; Siberian larch: RT = 1.1354, RC = 1.1354, and RS = 1.3046, $p < 0.05$). In 2003, the three resistance indicators of Siberian larch were still higher than those of Schrenk spruce (Schrenk spruce: RT = 1.028, RC = 1.033, and RS = 1.063; Siberian larch: RT = 1.05, RC = 1.041, and RS = 1.094, $p < 0.05$). However, the difference between the two tree species in terms of resistance to environmental disturbances gradually decreased after the temperature changed. The resistance indexes of both tree species declined in 2003 compared with 1981, especially those of Siberian larch.

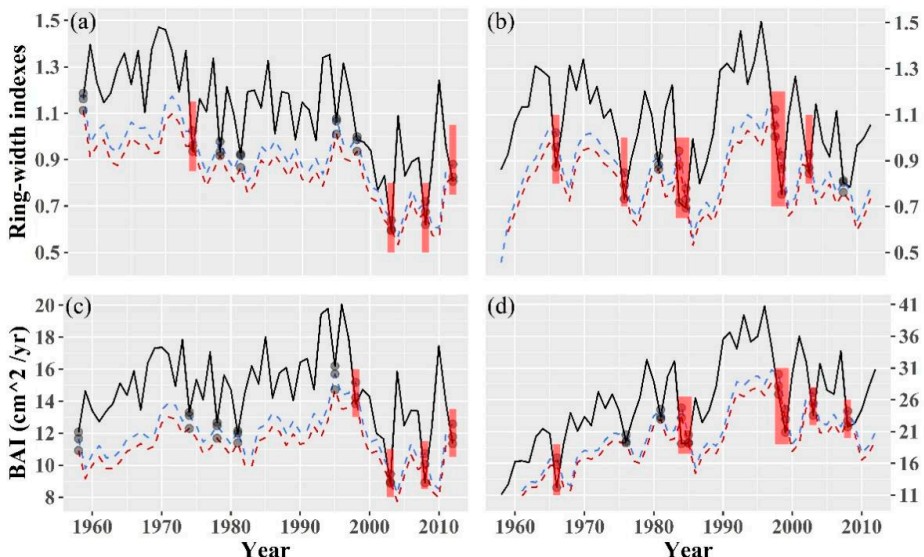

**Figure 6.** Variability of the standard chronologies for the two tree species and two narrow-ring horizontal lines. (**a**): Schrenk spruce, (**b**): Siberian larch. (**c**): BAI value of Schrenk spruce, (**d**): BAI value of Siberian larch. Solid black line: tree ring width of the standard chronologies, blue dotted line: 80% of the two-year moving average of the standard chronology, red dotted line: 75% of the two-year moving average of the standard chronology, gray dots: narrow years in the standard chronology with a value 20% less than the average of the previous two years, red rectangular box: narrow years in the standard chronology with a value 25% less than the average of the previous two years.

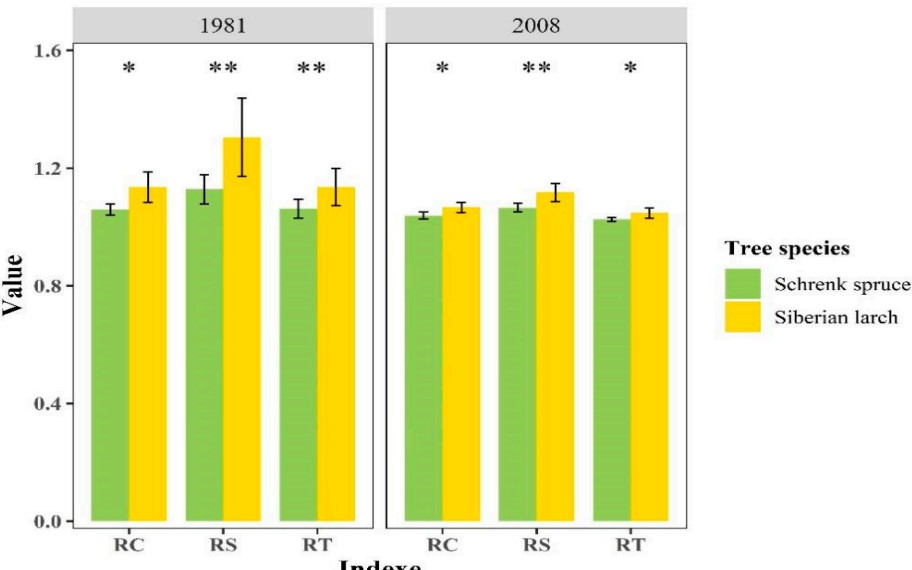

**Figure 7.** Histogram of the resistance indexes (RT: resistance, RC: recovery and RS: resilience) of the two species in 1981 and 2003. Asterisks indicate significance differences in the resistance index between Schrenk spruce and Siberian larch according to ANOVA, with *: significance level < 0.05, **: significance level < 0.01.

## 4. Discussion

### 4.1. Evaluation of the Responses of the Two Tree Species to Extreme Climate

The processes of tree leaf emergence and development, carbon accumulation, and xylem growth are all affected by climate, especially unstable extreme climate, which will have a more significant effect on tree-ring growth [44]. TRW includes abundant environmental change signals, which are direct and clear data that represent the responses of tree

radial growth to climate change [18]. Therefore, TRW can be used to accurately analyze the impact of climate change on forest ecosystems and evaluate the ecological response of tree growth to climate change.

Since the end of the last century, the trend of global warming has been obvious. The eastern Tianshan Mountains, located in the middle latitude of the Northern Hemisphere and central Asia, has a particularly significant interannual variation, especially the trend of the temperature rising and the abrupt change of the most extreme climate index in the 1980s (Figure 3; Table 4). After the temperature changed, the extreme warmth indexes (TXx, TX90P, TX10P, and SU25) and the minimum temperature indexes reflecting extreme cold (TNx and TNn) showed clear increasing trends, and the negative responses of Schrenk spruce and Siberian larch to those climate factors significantly strengthened (Figures 3 and 4). Under natural conditions, extreme high-temperature events are often accompanied by decreasing precipitation and increasing vapor pressure deficit (VPD), which further induce drought and aggravate drought stress on tree growth [1,45,46]. Extreme high temperature and drought damage the hydraulic structure of trees by cavitating the xylem and increasing duct embolism and can also cause tree "carbon starvation" by stimulating the closing of the stomata and inhibiting photosynthesis [47,48]. Decreasing trends could extend the growing season of plants (Figure 3). However, the probability of trees suffering from freezing damage has also increased with the extension of the growing season [33,49]. Specifically, trees in high-latitude areas have a narrower safety threshold for frost and greater risks than those in low-latitude areas [11]. Moreover, increasing temperatures before the growing season would increase the melting of snow and evaporation, which would aggravate the restriction of subsequent tree radial growth by limited water availability [19,50]. In addition, an increase in minimum temperature in the winter could reduce the overwintering mortality of pathogens and pests, thereby increasing the probability of pests and diseases in the coming year and ultimately affecting the radial growth of trees [48].

We also detected large differences in the times and types of responses of Schrenk spruce and Siberian larch to extreme climatic factors. After the temperature change during 1986–2012, the responses of Schrenk spruce to the extreme temperature indicators in the autumn of the previous year and the spring and summer of the current year became more significant (Figure 5). However, Siberian larch was more restricted by the extreme cold temperature indexes and extreme precipitation in the winter of the previous year and climatic factors in the autumn of the current year (Figure 5). Compared with Siberian larch, which is drought-tolerant and low-nutrient-tolerant, Schrenk spruce prefers a more humid and richer growth environment [51,52]. Therefore, Schrenk spruce is more susceptible to the obvious impact of climate change under significant increases in temperature and the frequency of extreme weather events (high temperature, drought, and frost) [53]. Moreover, Schrenk spruce, with shallow roots, is more susceptible to drought restrictions during the growing season due to its limited capacity for water storage and access to deep soil water resources [54]. In addition, evergreen Schrenk spruce was more significantly affected by climatic factors at the end of the growing season of the previous year than was deciduous Siberian larch because evergreen tree species continue to photosynthesize at the end of the growing season in order to maintain aboveground biomass [55,56]. However, the radial growth of Siberian larch is very sensitive to extreme precipitation variation in winter (Figure 5). The precipitation in winter is mostly in the form of snow in high-altitude areas at mid-to-high latitudes. Evergreen Schrenk spruce still retains a large number of branches and leaves in winter, and the amount of snow under the forest is correspondingly reduced since its canopy can intercept most of the snow. However, the amount of snow under larch forest is larger, and it takes longer for the snow to melt, which can delay the initiation of tree growth [57–59].

*4.2. Comparison of the Resistances of the Two Tree Species to Extreme Climate*

The frequency of narrow rings in the STDs and the resistance, recovery, and resilience of Schrenk spruce and Siberian larch were significantly different when the trees faced environmental disturbances, according to the occurrence of narrow rings and the resistance indicators reflecting radial growth (Figure 7). The narrow-ring years appearing in the STD of Schrenk spruce were all low-value years (1974, 1978, 2003, 2008, 2012, etc.), while Siberian larch was prone to consecutive low-value years. Schrenk spruce was more susceptible to occasional environmental disturbances, but the impact was relatively weak, while Siberian larch was more susceptible to sustained strong environmental stress, and the impact was relatively strong (Figure 6). High-frequency environmental interference might cause Schrenk spruce to develop strong tolerance in the face of subsequent interference [44]. Therefore, evergreen tree species and deciduous tree species may adopt different biomass allocation strategies under environmental stress conditions, with evergreen trees potentially choosing to preserve more aboveground biomass in order to maintain growth but larch choosing to abandon its leaves and branches and increase carbohydrate reserves in the rhizomes for subsequent recovery [60,61]. In addition, many studies have found that the growth rate of deciduous tree species is generally higher than that of evergreen tree species, and extreme climate events have more sudden and profound effects on deciduous tree species [62–64].

The radial growth of Schrenk spruce and Siberian larch was significantly lower in 1981 and 2003 than in other years (Figure 6). We first checked the historical data and ruled out the occurrence of biological interference in the two years [65]. For 1981, the warm autumn of the previous year may have allowed the trees to remain more active at the end of the growing season, consume more photosynthetic nutrients, and reduce the resource reserves for growth in the coming year. Therefore, the spring drought in 1981 (C3–C7) further aggravated the effect of water stress on tree growth during the growing season [19]. Before the abrupt climate change, Schrenk spruce was obviously restricted by the temperatures of the previous autumn and the spring of the current year, while Siberian larch was less restricted by the climate during these periods (Figures 4 and 5). Therefore, Schrenk spruce was more constrained by drought caused by high temperature and thus exhibited lower resistance and resilience in 1981. In 2003, the temperature was low throughout the year, especially in spring, and the precipitation in the previous winter increased significantly (Figure 2). The large amount of snow in winter and the low temperature at the beginning of the growing season slowed the increase in soil temperature, affected the emergence of new leaves, delayed growth, and shortened the growth period, thereby affecting the radial growth of trees [59,66,67]. After the abrupt temperature change, Siberian larch exhibited a significant negative correlation with the precipitation of the previous winter (Figure 5). Evergreen coniferous species usually use nonstructural carbohydrates from old leaves for bud burst, and it takes longer for new evergreen leaves to emerge [68,69]. However, the bud burst period of deciduous trees is earlier and is more affected by the low temperatures and frost in spring because these trees rely on the NSC from stems and branches for leaf development [70,71]. Therefore, the low temperature in the spring of 2003 had a significant impact on the early budding and radial growth of Siberian larch. In addition, the fluctuations in climate in 2003 were more obvious than those in 1981. Studies have shown that the resistance of trees to climate change is closely related to the intensity of extreme climate events [28]. This might be one of the important reasons why the resistance difference between Schrenk spruce and Siberian larch decreased. On two occasional climate disturbance events, Siberian larch showed better resistance and resilience, so we believe that Siberian larch may be better at adapting to extreme environmental events in the context of climate warming.

The frequency and intensity of extreme climate events have increased significantly since the beginning of the 21st century. The occurrence of extreme climate events has significantly disrupted the growth, death, and regeneration of trees, thereby affecting the structure and function of entire forest ecosystems [72]. Especially in arid and semiarid

regions, trees have shown significant declines in growth rates and increased mortality due to drought and climate warming [73]. Evergreen Schrenk spruce and deciduous Siberian larch are dominant species in the eastern Tianshan Mountains. The adaptations of these two tree species to climate change exhibit obvious variation due to significant differences in genetic physiology and ecological thresholds between the species [74]. According to our research results, Schrenk spruce is more sensitive to climate and more affected by extreme climatic changes, while Siberian larch exhibits better resistance and a greater recovery ability when faced with the same climate disturbances. Moreover, extreme climate events do not occur in isolation. For example, extreme high temperature and heat wave events are often accompanied by drought, extreme precipitation events are often accompanied by hurricanes or storms, and extreme low-temperature events may also be accompanied by cold waves and hail [75]. The response of trees to extreme climate events is also affected by factors such as tree species and the type and intensity of the extreme event [76]. Therefore, we should pay more attention to the effects of extreme climate on tree growth and the adaptation of tree growth to climate change in ecologically fragile and sensitive areas. We also need to conduct more research to assess the adaptability of different tree species to extreme climatic events under future climate change.

## 5. Conclusions

The growth of trees and the structure and stability of forest ecosystems have been disturbed by the more frequent occurrence of extreme climatic events with global warming. Schrenk spruce and Siberian larch are the most dominant mountain conifers in the arid and semiarid regions of central Asia. Therefore, studying their growth patterns and responses to extreme climate in this region can provide importance references. Our research results showed that Schrenk spruce and Siberian larch had strong responses to 27 extreme climatic indexes. Specifically, the correlations between characteristics of the two tree species and the extreme climatic indexes increased after the temperature changed. This confirmed that the restrictive effect of extreme climate on forest ecosystems gradually increased with the intensification of climate change. Therefore, it is necessary to further strengthen the dynamic monitoring of extreme climate and forest ecosystems and research on their impact mechanisms. In addition, our study also revealed that the evergreen spruce species was more susceptible to extreme climate, while the deciduous species (Siberian larch) had stronger resistance and a greater recovery ability after being disturbed by the extreme environment. Therefore, forest ecosystems should be protected and managed more specifically according to differences in the responses and resistances of tree species to extreme climatic events under future climate change.

**Author Contributions:** Methodology: L.J. and X.W.; software: X.W.; validation: L.J. and X.W.; formal analysis: L.J. and X.W.; investigation: L.J., X.W., X.L., C.Q. and R.X.; resources: L.J. and X.L.; data curation: L.J. and X.W.; writing—original draft: L.J. and X.W.; writing—review and editing: L.J., X.L. and X.W.; visualisation: X.L., C.Q., R.X. and D.D.; supervision: L.J., X.L., D.D., C.Q. and R.X.; project administration: L.J. and X.W.; funding acquisition: L.J. All authors have read and agreed to the published version of the manuscript.

**Funding:** This research was supported by Natural Science Foundation of Gansu (No. 21JR7RA111), CAS "Light of West China" Program (2020XBZG-XBQNXZ-A), and the 2022 Major Scientific Research Project Cultivation Plan of Northwest Normal University (WNU-LKZD2022-04).

**Data Availability Statement:** The data set analyzed during the current research period is provided in China Meteorological Data Network. Other data sets analyzed during the current research period may be obtained from the corresponding author upon reasonable request.

**Acknowledgments:** We also thank the anonymous referees for helpful comments on the manuscript.

**Conflicts of Interest:** The authors declare that they have no known competing financial interests or personal relationships that could have appeared to influence the work reported in this paper.

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
