# Peer review of "Ecological Adaptation of Two Dominant Conifer Species to Extreme Climate in the Tianshan Mountains"

_forests, doi:10.3390/f14071434_

Round 1

Reviewer 1 Report

The Indroduction is well-written and give some useful insights into the subject. Some aspects that could be added are the following.

Line 40. It would be useful to mention some recent developed tools for climate change simulation. The most modern tools for future climate projections are Regional Climate models (RCMs). These models have been tested over forested areas with complex terrain and should be integrated into forest management and adaption planning (https://scibulcom.net/en/article/dKhTT3KVYdZNA4wZPbkR, https://doi.org/10.1007/s10980-023-01678-y).

Line 90. Clearly state the research gap and the usefulness of the research output.

Line 111. Increase the fonts in figure 2

Line 181. Are these changes statistical significant? At which level?

Line 247. Please reduce the number of decimals to 4 in all columns.

Figure 5. Increase the Y – axis of the figure. It is not readable!

Author Response

Review 1

Line 40. It would be useful to mention some recent developed tools for climate change simulation. The most modern tools for future climate projections are Regional Climate models (RCMs). These models have been tested over forested areas with complex terrain and should be integrated into forest management and adaption planning. 

(https://scibulcom.net/en/article/dKhTT3KVYdZNA4wZPbkR,https://doi.org/10.1007/s10980-023-01678-y).

Answer: Thank you very much for your reasonable suggestion! The content of this section is further supplemented below. “Therefore, it is particularly important to study the response relationship between extreme climate indicators and tree growth in order to explore tree growth strategies and formulate management and protection measures under future climate change. At present, in order to further strengthen forest management and protection and scientific planning, Climate change modeling tools such as the Forest Landscape Model (RCMs), which includes the effects of climate and management on forest dynamics, have been used to predict the layout of forest growth under future climate impacts, especially in forest areas with complex terrain.”(Line41-49)

Line 90. Clearly state the research gap and the usefulness of the research output.

Answer: Thank you very much! The research content and significance mentioned here are further supplemented as follows. “In these studies, the reconstruction of climate indicators using accurate long-sequence tree-ring data is conducive to revealing long-term climate dynamics in different regions of the world, and exploring the response relationship between tree growth and climate is conducive to predicting forest growth patterns under future climate change.”(Line98-102)

Line 111. Increase the fonts in figure 2

Answer: Thank you very much for your scientific suggestion! I have made appropriate adjustments to the font in the picture.(Line121)

Line 181. Are these changes statistical significant? At which level?

Answer: Thank you very much for your suggestions here! These changes are statistically significant, and the interannual change rates and abrupt change of 28 climate indexes in 1958-2012 are mainly presented here. “Pettitt method” is a statistical method used to detect the mutation points in time series data and determine whether the mutation is significant. This method is based on the probability of finding the maximum mutation point in the time series, and is a non-parametric method. The significance level was p=0.01. p=0.05; p=0.001. The climate index in the figure passed the significance test, p<0.05.

Line 247. Please reduce the number of decimals to 4 in all columns.

Answer: Thanks! I have corrected the details in the table.(Line244)

Figure 5. Increase the Y-axis of the figure. It is not readable!

Answer: Thank you very much for your reasonable suggestion! The Y-axis coordinates in the image indicate different extreme climate indicators, and the font size of the coordinates has been adjusted to make them clear.(Line274)

Reviewer 2 Report

Global climate warming is one of the most significant challenges to humanity. It is generally recognized that forests can mitigate the effects of climate change. However, the forests themselves are also influenced by climatic factors. Conservation of the biodiversity of woody plants is becoming an increasingly urgent task. For these purposes, information is needed on the resistance of trees to various factors, as well as on the adaptive potential. Therefore, the topic of this paper is relevant.

The scientific novelty lies in the identification of the relationship between radial growth of the dominant tree species (Schrenk spruce and Siberian larch) and extreme climate factors in the eastern Tianshan Mountains, an area with more arid conditions.

The research results have both theoretical and applied significance. However, the authors missed this point. I can advise you to pay special attention to it in the paper. Then readers will be more informed and more interested.

The relevance of the study is well argued in the introduction. The description of the current state of the problem could be improved. The authors did not include studies published after 2020 in their analysis. The research objectives are clearly stated. It may be advisable to identify more clearly the target audience to whom the research findings will be most useful.

 The methodological approaches are described in detail. The authors have used methods appropriate to the task at hand. These methods have been repeatedly tested in research and are recognized as effective. The authors chose Pearson correlations as the main criterion for the strength of the relationships.

 The research findings are illustrated with informative and non-duplicative figures and tables. The paper contains 4 informative tables and 7 visual figures. The results are presented in a clear and unambiguous manner. The paper contains a lot of abbreviations that are hard to remember. This reduces the readability of the text and the interest in the study. I also lost the meaning of the text in some places because of this.  I advise authors to reduce the use of the abbreviation as much as possible.

 The conclusions drawn from the results are reasonable.  The paper will be of interest to a wide range of readers with a scientific interest in climate change and trees. Although English is not my first language, I read the paper with interest and had no difficulty understanding it. The paper is appropriate to the topic and level of Forests. Here you can pay attention to the significance of the research, the scope and limitations of the application of the research results.

The reference must be issued in accordance with the requirements of the Forests.

Author Response

Review 2

Global climate warming is one of the most significant challenges to humanity. It is generally recognized that forests can mitigate the effects of climate change. However, the forests themselves are also influenced by climatic factors. Conservation of the biodiversity of woody plants is becoming an increasingly urgent task. For these purposes, information is needed on the resistance of trees to various factors, as well as on the adaptive potential. Therefore, the topic of this paper is relevant.

The scientific novelty lies in the identification of the relationship between radial growth of the dominant tree species (Schrenk spruce and Siberian larch) and extreme climate factors in the eastern Tianshan Mountains, an area with more arid conditions.

The research results have both theoretical and applied significance. However, the authors missed this point. I can advise you to pay special attention to it in the paper. Then readers will be more informed and more interested.

The relevance of the study is well argued in the introduction. The description of the current state of the problem could be improved. The authors did not include studies published after 2020 in their analysis. The research objectives are clearly stated. It may be advisable to identify more clearly the target audience to whom the research findings will be most useful.

The methodological approaches are described in detail. The authors have used methods appropriate to the task at hand. These methods have been repeatedly tested in research and are recognized as effective. The authors chose Pearson correlations as the main criterion for the strength of the relationships.

The research findings are illustrated with informative and non-duplicative figures and tables. The paper contains 4 informative tables and 7 visual figures. The results are presented in a clear and unambiguous manner. The paper contains a lot of abbreviations that are hard to remember. This reduces the readability of the text and the interest in the study. I also lost the meaning of the text in some places because of this.I advise authors to reduce the use of the abbreviation as much as possible.

The conclusions drawn from the results are reasonable. The paper will be of interest to a wide range of readers with a scientific interest in climate change and trees. Although English is not my first language, I read the paper with interest and had no difficulty understanding it. The paper is appropriate to the topic and level of Forests. Here you can pay attention to the significance of the research, the scope and limitations of the application of the research results.

The reference must be issued in accordance with the requirements of the Forests.

Answer: Thank you very much for your affirmation of this research and some comments. Your comments fully explain the significance of this research and the rationality of the presentation of research results. At the same time, some of your suggestions are also very scientific and reasonable, which plays a very important role in improving the paper.

Most of the abbreviations in the article refer to the names of 27 extreme climate indicators, so they appear more frequently. However, according to your suggestion, the frequency of use of some abbreviations has been adjusted (Line382). study published after 2020 are also added to the introduction (Line 41-48 ).

Round 2

Reviewer 2 Report

The authors have responded to all my comments. I have no further comments.